# Perceptions of COVID-19 Vaccines: Protective Shields or Threatening Risks? A Descriptive Exploratory Study among the Italian Population

**DOI:** 10.3390/vaccines11030642

**Published:** 2023-03-13

**Authors:** Paola Boragno, Elena Fiabane, Irene Taino, Marina Maffoni, Valentina Sommovigo, Ilaria Setti, Paola Gabanelli

**Affiliations:** 1Istituti Clinici Scientifici Maugeri IRCCS, Psychology Unit of Pavia Institute, 27100 Pavia, Italy; 2Istituti Clinici Scientifici Maugeri IRCCS, Psychology Unit of Montescano Institute, 27040 Montescano, Italy; 3Department of Psychology, Faculty of Medicine and Psychology, Sapienza University of Rome, 00185 Roma, Italy; 4Unit of Applied Psychology, Department of Brain and Behavioural Sciences, University of Pavia, 27100 Pavia, Italy

**Keywords:** vaccine hesitancy, COVID-19, vaccines, ambivalence, safety, infodemic

## Abstract

Although several quantitative studies have explored vaccine hesitancy, qualitative research on the factors underlying attitudes toward vaccination is still lacking. To fill this gap, this study aimed to investigate the general perceptions of COVID-19 vaccines among the Italian population with a qualitative approach. The sample included 700 Italian participants who completed an online survey. Open questions underwent a descriptive analysis for unveiling meaning categories, while differences in the prevalence of categories were calculated using chi-square or Fisher’s exact tests. Vaccination was associated with the following seven main themes: ‘safety’, ‘healthcare’, ‘vaccine delivery’, ‘progress’, ‘ambivalence’, ‘mistrust’, and ‘ethics’. Vaccinated individuals more frequently reported words related to the safety theme (χ^2^ = 46.7, *p* < 0.001), while unvaccinated individuals more frequently reported words related to mistrust (χ^2^ = 123, *p* < 0.001) and ambivalence (χ^2^ = 48.3, *p* < 0.001) themes. Working in the healthcare sector and being younger than 40 years affected the general perceptions of vaccination in terms of pro-vaccine attitudes. Unvaccinated individuals were more affected by the negative experiences of their acquaintances and manifested more distrust of scientific researchers, doctors, and pharmaceutical companies than vaccinated individuals. These findings suggest promoting collaborative efforts of governments, health policymakers, and media sources, including social media companies, in order to deal with cognitions and emotions supporting vaccine hesitancy.

## 1. Introduction

Over the past few centuries, vaccination programs have led to the global eradication of numerous infectious diseases, reducing mortality and morbidity rates [1]. Nevertheless, infectious diseases are an ever-present threat to humans. Therefore, high vaccination coverage rates are necessary to reduce the spread of life-threatening diseases, helping people of all ages live longer and healthier [2]. Recently, on 30 January 2020, the World Health Organization (WHO) declared the COVID-19 outbreak a public health emergency of international concern [3]. With the lack of specific therapeutic interventions and drugs, vaccination was the most effective strategy for mitigating and suppressing complications associated with the disease [4,5]. Despite the benefits of vaccination, some individuals were reluctant to vaccinate, obstructing the reach of herd immunity, which is the key to safeguarding humanity [6]. In this regard, the WHO defines vaccine hesitancy as a ‘delay in acceptance or refusal of safe vaccines despite the availability of vaccine’ [7]. Vaccine hesitancy arises against all types of vaccines worldwide [8,9,10]. The WHO’s Strategic Advisory Group of Experts on Immunization (SAGE) has defined vaccine hesitancy as a multidimensional and complex phenomenon, varying by time, place, vaccine, subgroup, and person [11]. Indeed, many researchers have shown that the mere availability of a vaccine does not equal its acceptance [12,13,14]. In recent years, global concerns about public acceptance of vaccines have increased [15,16]. The scientific literature has shown that the decision to vaccinate may be affected by emotional, cultural, social, spiritual, logistical, political, and cognitive factors [17,18,19]. Therefore, a better understanding of global opinions, concerns, and beliefs is urgent and essential for effective public health communication.

Factors associated with hesitancy toward COVID-19 vaccines are similar to those associated with hesitancy toward other vaccines and have been categorized into groups related to vaccine characteristics, political factors, and individual characteristics [20]. For example, previous studies have found that hesitant attitudes toward vaccines were associated with younger age [21,22,23], female gender [21,23], lower education [21,22,23], lower income [23], and non-liberal political views [22]. Among the other factors associated with individuals’ willingness to be vaccinated, research has identified the trust in various institutions, attitudes, and beliefs about vaccines and related benefits, in addition to attitudes and perceptions toward healthcare workers [24,25,26]. Indeed, many studies conducted during the recent pandemic have shown that hesitant or anti-vaccine individuals reported mistrust in government and health authorities as the reason underlying their hesitancy toward or refusal of vaccination [27]. Other studies have instead reported that attitudes and perceptions of healthcare personnel toward vaccination (e.g., skepticism, noncompliance, concerns regarding safety, etc.) are the main factors linked to patients’ vaccine acceptance and hesitancy [28,29]. Therefore, examining the vaccination acceptance levels and opinions among healthcare workers would help policymakers, practitioners, and health authorities to design appropriate strategies and interventions to reduce vaccine hesitancy. Although, to date, different studies have examined vaccine hesitancy/adherence [21,22,23,26], to the best of our knowledge, no qualitative studies have investigated factors influencing perceptions of vaccination in a large sample of the general population. To fill this gap, we have conducted a cross-sectional qualitative study on a large convenience sample of the Italian population with a triple objective: (a) to qualitatively explore general population perceptions toward vaccination; (b) to evaluate the impact of three relevant factors (i.e., COVID-19 vaccination, working as a healthcare professional, and age) on vaccination perceptions; (c) to preliminarily investigate risk perceptions and factors influencing attitudes toward vaccination among vaccinated and unvaccinated individuals.

## 2. Materials and Methods

### 2.1. Participants and Procedure

This descriptive qualitative investigation was conducted between March and April 2022 in Northern Italy as a part of a larger research project investigating vaccine hesitancy [17]. Participants completed an online survey using a spreadsheet on Google Sheets. Participants were contacted through mailing lists and social networks or using a snowball sampling procedure (i.e., each participant invited their acquaintances to complete the online survey). The survey explored opinions associated with the ‘vaccine’ word, perceptions of vaccine-related risks, and personal attitudes toward vaccination. The inclusion criteria for participants in this study were: (a) native Italian speakers; (b) at least 18 years of age. This study was approved by the Ethics Committee of the University of Pavia, Italy. Before participating, each subject was required to sign and read an informed consent form explaining the objectives, protocol, and data storage methods. Participants were also provided with information about the anonymity, confidentiality of responses, and their right to interrupt the fulfillment of the questionnaire at any time without any explanation. The total sample was composed of 700 Italian individuals with a mean age of 41.23 (SD = 15.66). Most of them were women (74.1%) who did not work in the healthcare sector (77.7%) and were vaccinated against COVID-19 (92.14%).

### 2.2. Measurements

A questionnaire was developed ad hoc to investigate individual perceptions toward vaccination (see [17] for a detailed description of the entire questionnaire). Firstly, an open-ended question was administered to qualitatively investigate immediate and general perceptions toward vaccination (i.e., ‘What image/word does come to mind when you think of vaccines?’). Two items with multiple-choice were used to examine risk perceptions (i.e., ‘Why do you think getting vaccinated may be risky?’; a sample answer for this question is ‘Personal negative experience’) and factors influencing attitudes towards vaccination (i.e., ‘Which of the following variables do you think impact more on your attitude toward vaccination?’; a sample answer for this question is ‘Mode of administration (needle versus spray)’). Finally, individuals were invited to answer a dichotomous question (1 = Yes, 2 = No) regarding whether they were vaccinated against COVID-19 (i.e., ‘I have been vaccinated against COVID-19 ‘) and give some socio-demographic information (i.e., age, gender, and whether the participant is employed as a healthcare professional). The administered survey is described in Table 1.

### 2.3. Statistical Analyses

Firstly, we calculated the descriptive statistics of the sample. Specifically, we examined participants’ answers to b and c questions (see Table 1). The presence of possible differences in the answers given to these two questions between individuals who were vaccinated against COVID-19 and those who were unvaccinated were analyzed using the chi-square test of independence with two tails (*p* < 0.05). Then, we analyzed the terms collected in the open question in order to quantify them and identify the main themes associated with vaccines [30]. To this end, we conducted a descriptive analysis to identify categories through a bottom-up approach based on meaning. Adjectives and verbs were changed to nouns, and plural forms were turned into singular ones if no changes in meaning occurred. The terms considered synonyms in the Collins Thesaurus were combined, and the most often mentioned terms by the participants were chosen. Typing errors were corrected, and semantically irrelevant terms were deleted. Then, the authors determined conceptual themes. All themes were discussed among authors (P.B., I.T., M.M.) until a consensus was achieved. The other authors (P.G., E.F., V.S., I.S.) supervised the process and provided a final evaluation of the identified meaning categories. After identifying the main categories, we analyzed differences in the frequency of themes between individuals who were vaccinated against COVID-19 and those who were not, between healthcare workers and non-healthcare workers, and between participants aged 40 or younger and participants aged over 40. Differences in the prevalence of categories were calculated using the chi-square test of independence and Fisher’s exact test with two tails (*p* < 0.05).

## 3. Results

### 3.1. Prevalence of Themes in the Total Sample

Seven themes related to the ‘vaccine’ word were extracted (see Table 2). Of the total sample, 62.7% reported words included in the ‘safety’ theme, followed by ‘healthcare’ (8.9%), ‘vaccine delivery’ (7.9%), ‘progress’ (6%), ‘ambivalence’ (5.3%), ‘mistrust’ (4.6%), and ‘ethics’ (3.1%) themes.

### 3.2. Differences in the Prevalence of Themes between COVID-19-Vaccinated and Unvaccinated Individuals

Vaccinated individuals reported words related to the ‘safety’ theme statistically significantly more frequently (66.41%) than unvaccinated individuals (20%) (χ^2^(1, *N =* 700) = 46.7, *p* < 0.001). Unvaccinated individuals (25.45%) reported words related to the ‘ambivalence’ theme more frequently than those who were vaccinated (3.58%) (χ^2^(1, *N =* 700) = 48.3, *p* < 0.001).

A statistically significantly higher percentage of unvaccinated individuals (34.5%) mentioned words related to the ‘mistrust’ theme in comparison with those who were vaccinated (2.02%) (χ^2^(1, *N =* 700) = 123, *p* < 0.001). There were no statistically significant differences in the prevalence of ‘healthcare’ (χ^2^(1, *N =* 700) = 2.37, *p* = 0.124) and ‘vaccine delivery’ themes (Fisher’s test; *p* = 0.30) between vaccinated and unvaccinated groups. Only the vaccinated group reported words, phrases, or pictures related to the ‘progress’ (6.53%) and ‘ethics’ (3.25%) themes. Statistically significant comparisons are summarized in Figure 1.

### 3.3. Differences in the Prevalence of Themes between Healthcare Workers and Non-Healthcare Workers

Healthcare workers mentioned words related to the ‘safety’ theme statistically significantly more frequently (77.56%) than non-healthcare workers (58.49%) (χ^2^(1, *N =* 700) = 18.9, *p* < 0.001). Healthcare workers reported words related to the ‘healthcare’ theme statistically significantly more frequently (10.33%) than non-healthcare workers (3.84%) (χ^2^(1, *N =* 700) = 6.30, *p* < 0.05). A statistically significantly higher percentage of non-healthcare workers (5.53%) mentioned words related to the ‘mistrust’ theme than healthcare workers (2.02%) (χ^2^(1, *N =* 700) = 5.01, *p* < 0.05). There were no statistically significant differences in the prevalence of ‘progress’ (χ^2^(1, *N =* 700) = 0.83, *p* = 0.362), ‘vaccine delivery’ (χ^2^(1, *N =* 700) = 1.23, *p* = 0.267), ‘ambivalence’ (Fisher’s test; *p* = 0.104), and ‘ethics’ (Fisher’s test; *p* = 0.798) themes between the two groups. Statistically significant comparisons are reported in Figure 1.

### 3.4. Differences in the Prevalence of Themes between Individuals Aged 40 or Younger and Individuals Aged over 40

Participants aged 40 or younger mentioned words related to the ‘vaccine delivery’ theme statistically significantly more frequently (13.46%) than participants aged over 40 years old (2.56%) (χ^2^(1, *N =* 700) = 27.7, *p* < 0.001). A statistically significantly higher percentage of participants aged over 40 years old (7.1%) mentioned words related to the ‘mistrust’ theme in comparison with participants aged 40 or younger (2.02%) (χ^2^(1, *N =* 700) = 0.225, *p* = 0.635). There were no statistically significant differences in the prevalence of ‘safety’ (χ^2^(1, *N =* 700)= 1.62, *p* = 0.201), ‘healthcare’ (χ^2^(1, *N =* 700) = 0.11, *p* = 0.736), ‘progress’ (χ^2^(1, *N =* 700) = 1.03, *p* = 0.311), ‘ambivalence’ (χ^2^(1, *N =* 700) = 3.26, *p* = 0.071), and ‘ethics’ (χ^2^(1, *N =* 700) = 0.225, *p* = 0.635) themes between the two groups. Statistically significant differences are summarized in Figure 1.

### 3.5. Perception of Risk and Attitudes towards Vaccination: Differences between Vaccinated against COVID-19 and Unvaccinated Individuals

#### 3.5.1. Answers to the Question ‘Why Do you Think Getting Vaccinated May Be Risky?’

A statistically significant higher percentage of vaccinated individuals (49.15%) believed that getting vaccinated was not risky in comparison with those who were not vaccinated (3.64%) (χ^2^(1, *N =* 700) = 42.32, *p* < 0.001). A statistically significant higher percentage of unvaccinated individuals (47.27%) considered vaccination a risk due to negative experiences of relatives and/or friends than the vaccinated group (8.53%) (χ^2^(1, *N =* 700) = 74.35, *p* < 0.001). Unvaccinated individuals (25.45%) believed that vaccination was risky more frequently (25.45%) than vaccinated individuals (12.40%) (χ^2^(1, *N =* 700) = 7.43, *p* < 0.05) due to the negative experiences reported by their acquaintances. Unvaccinated individuals considered vaccination a risk to medical opinion statistically significantly more frequently (38.18%) than vaccinated individuals (16.12%) (χ^2^(1, *N =* 700) = 16.81, *p* < 0.001). There were no statistically significant differences in the answers to ‘negative personal experiences’ (χ^2^(1, *N =* 700) = 3.36, *p* = 0.067) and ‘media (TV, Internet, etc.), press’ (χ^2^(1, *N =* 700) = 3.08, *p* = 0.079) questions between the two groups. A statistically significant higher percentage of unvaccinated individuals (52.73%) selected the ‘scientific journal’ option in comparison with the vaccinated group (16.59%) (χ^2^(1, *N =* 700) = 42.28, *p* < 0.001). Statistically significant differences are shown in Table 3.

#### 3.5.2. Answers to the Question ‘Which of the following Variables Do You Think Impact More on Your Attitudes toward Vaccination?’

Compared with vaccinated individuals (31.01%), a statistically significant higher percentage of unvaccinated individuals (76.36%) reported that the variable type of vaccine affected their attitude toward vaccination (χ^2^(1, *N =* 700) = 49.09, *p* < 0.001). There were no statistically significant differences in ‘mode of administration (needle vs. spray)’ (χ^2^(1, *N =* 700) = 1.93, *p* = 0.275) and ‘opinions of friends or relatives’ (χ^2^(1, *N =* 700) = 2.79, *p* = 0.094) answers between these two groups. Compared with the unvaccinated group (1.82%), a statistically significant higher percentage of vaccinated individuals (15.50%) reported that trust in pharmaceutical companies affected their attitude towards vaccination (χ^2^(1, *N =* 700) = 7.69, *p* < 0.05). Compared with vaccinated individuals (10.85%), a statically significantly higher percentage of unvaccinated individuals (78.18%) reported that distrust of pharmaceutical companies affected their attitude toward vaccination (χ^2^(1, *N =* 700) = 169.71, *p* < 0.001). Vaccinated individuals reported that trust in healthcare workers and doctors affected their attitudes toward vaccination statistically significantly more frequently (40%) than unvaccinated individuals (3.64%) (χ^2^(1, *N =* 700) = 28.70, *p* < 0.001). Unvaccinated individuals reported that distrust of healthcare workers and doctors affected their attitude toward vaccination statistically significantly more frequently (38.18%) than vaccinated individuals (2.79%) (χ^2^(1, *N =* 700) = 120.65, *p* < 0.001). The latter reported that trust in scientific research affected their attitudes toward vaccination statistically significantly more frequently (84.81%) than unvaccinated individuals (16.36%) (χ^2^(1, *N =* 700) = 145.29, *p* < 0.001). Unvaccinated individuals reported that distrust of scientific research affected their attitudes toward vaccination statistically significantly more frequently (21.82%) than vaccinated individuals (3.26%) (χ^2^(1, *N =* 700) = 38.87, *p* < 0.001). Unvaccinated individuals reported that mass media, newspapers, and magazines affected their attitudes toward vaccination statistically significantly more frequently (21.82%) than those who were vaccinated (6.05%) (χ^2^(1, *N =* 700) = 18.66, *p* < 0.001).

## 4. Discussion

Several quantitative studies have shown that vaccine hesitancy is a common phenomenon globally, with a wide variability in the factors associated with the refusal of vaccination acceptance [31]. Compared to the large body of quantitative research on vaccine hesitancy, fewer qualitative studies have analyzed this phenomenon, which is a gap that needs to be addressed as qualitative methods are crucial to reaching an in-depth understanding of this topic [32]. Thus, qualitatively investigating which factors contribute to vaccine hesitancy can assist in the planning of tailored communication campaigns and fighting public health threats. To date, most qualitative studies in the scientific literature have investigated opinions about vaccines through semi-structured interviews with open-ended responses or content analysis posted on social media [18,19,32,33,34,35]. Conversely, there is a paucity of qualitative research on free associations related to the vaccine topic provided by vaccinated versus unvaccinated people. To fill this gap, we asked our research participants to indicate which words or images came to mind when they thought of vaccines.

Our findings showed that most participants (62.7%) referred to words concerning the ‘safety’ theme, suggesting that vaccination was perceived as a protective shield against diseases and a highly effective treatment tool. This result is consistent with the previous literature, demonstrating that most of the general population considered vaccination an effective and necessary means to protect against a disease perceived as an uncertain, uncontrollable, and dangerous risk [11,17,35].

The vaccine word was also associated with the ‘healthcare’ (8.9%) and ‘vaccine delivery’ (7.9%) themes. On the one hand, regarding the ‘healthcare’ theme, several participants mentioned words such as ‘COVID-19’, ‘pandemic’, or ‘virus’. Even though the original question intended to explore perceptions of vaccination in general, these collected words denoted an overlapping between the COVID-19 vaccine and vaccination in general, probably due to the broad worldwide impact and the timing of this research. On the other hand, regarding the ‘vaccine delivery’ theme, words such as ‘needle’, ‘puncture’, and ‘syringe’ were cited, suggesting that vaccines may also be regarded as intrusive and painful. These results are of particular interest since we previously found that the route of administration can affect vaccination uptake [17].

The ‘progress’ theme emerged from 6% of answers related to trust in scientific research and science (e.g., ‘evolution’). Interestingly, all individuals included in this category were vaccinated against COVID-19, confirming the results by Barattucci and colleagues (2012) [36]. Thus, the authors have shown that trust in science plays a crucial role in predicting vaccination intention, mediating the relationship between individual factors (i.e., fear of COVID-19, and subjective norms) and vaccination intention [36]. Thus, trust in science can be viewed as a key factor contributing to individuals’ compliance with government guidelines (e.g., vaccination) [36]. Unlike the ‘progress’ theme, the ‘ambivalence’ theme, which was reported by 5.3% of the sample (e.g., ’uncertainty’), was strictly focused on vaccine-related uncertainties and fears, which might negatively impact vaccination intention as shown in the study of Perrone and colleagues (2023) [19]. Thus, the results of the latter study, which was conducted adopting an interpretative descriptive approach, indicated that emotional factors (e.g., fear of vaccine side effects and lack of control) affect vaccination adherence [19]. Additionally, a quantitative study demonstrated that the perception of low control over events and high levels of intolerance of uncertainty can increase doubts and concerns, leading people to delay and postpone vaccine decisions [17]. Moreover, in this study, we showed that general anti-vaccination opinions were strictly connected with the ‘mistrust’ theme. The few respondents who reported this theme (4.6%) considered vaccination a fraud and an obligation, mistrusting agencies that monitor vaccine development and distribution, healthcare workers who deliver vaccines, and pharmaceutical companies dealing with vaccine production [37]. Finally, a small portion of the sample (3.1%) reported words belonging to the ethical domain. The ‘ethics’ theme (e.g., ‘an act of responsibility’) emerged exclusively from subjects vaccinated against COVID-19, suggesting that, in this study, the concept of vaccine as a moral duty was internalized by vaccinated people only. This result is consistent with what was found in the work of Burke and colleagues (2021) [38]. The authors demonstrated that collectivist and altruistic beliefs positively affect vaccination uptake [38].

This study also aimed to examine differences in the prevalence of these themes regarding three factors that the literature has previously shown to be involved in vaccine hesitancy, namely (1) having been (un-)vaccinated against COVID-19; (2) working in the healthcare sector; and (3) age [21,22,23,39]. In this regard, our findings showed a higher prevalence of the ‘safety’ theme among vaccinated people than unvaccinated people. Such perceptions and attitudes help avoid cognitive dissonance and experience consistency, as supported by cognitive consistency theory [40,41]. Indeed, individuals who have already received the vaccination are unlikely to doubt the usefulness or safety of this choice because this could lead them to experience an unpleasant state of cognitive dissonance [40,41]. Conversely, unvaccinated individuals showed higher levels of the ‘mistrust’ and ‘ambivalence’ themes compared to those who were vaccinated. Unvaccinated people gave an image of vaccination that reflected the lack of trust in such a preventive measure, doubting vaccine effectiveness and worrying about possible side effects. Additionally, some of them even considered vaccination risky for their health. A recent review identified the lack of confidence in vaccine safety as one of the main reasons for low vaccination acceptance rates in Italy [10]. Likewise, Cristea and colleagues (2022) interpreted vaccine hesitancy as an expression of concern about vaccination safety and distrust in the authorities among the Romanian population [42]. In this regard, vaccination refusal could be considered a kind of ‘political stance’ to fight obligations laid down by political institutions, which could be perceived as restrictive measures of personal freedoms.

The results of this study also showed that healthcare workers more frequently reported words related to the ‘trust’ and ‘safety’ themes when compared to non-healthcare workers. In this regard, previous studies (e.g., [43]) demonstrated that hospital workers are more confident in and willing to be vaccinated than non-healthcare workers. This could suggest that, among healthcare workers, the use of words and images regarding safeguarding reflects a view of vaccines that remains anchored in their scientific education. This result is encouraging as healthcare workers play a central role in communicating vaccine importance and safety, which may ultimately encourage the wider population to receive vaccinations. Finally, the results of age group comparison indicated that individuals aged over 40 years reported words related to the ‘mistrust’ theme more frequently than those aged 40 or younger. This finding contradicts the results of previous studies showing that older age was associated with low vaccine hesitancy [21,22,23]. A plausible explanation is that, during the COVID-19 pandemic, young adults viewed vaccination as an opportunity to return to their social life and enjoyable activities more than older adults [44].

Finally, this study investigated whether there were differences in risk perceptions and factors influencing attitudes toward vaccination between individuals who were vaccinated against COVID-19 and those who were not. Our results showed that risk perceptions were statistically significantly lower among vaccinated individuals. This result confirms that vaccine refusal or delay results from an imbalance between the perception of the vaccine-related risk and benefits associated with vaccination: individuals who weigh the risks related to vaccination higher than its benefits will tend to refuse or delay some or all vaccines [33,45,46]. The health belief model supports this finding by highlighting that perceptions of vaccine efficacy and concerns about vaccine-related side effects may hinder subjects from getting vaccinated [47]. Furthermore, in this study, a higher percentage of unvaccinated individuals believed that getting vaccinated was risky due to negative experiences reported by their relatives, friends, and acquaintances. These results are consistent with those from the study of Facciolà and colleagues (2019) [1]. The authors confirmed that unfavorable opinions about vaccination seem to be conditioned on direct or indirect knowledge of people harmed by vaccines. In addition, Slovic (1987) demonstrated that individuals often rely on information consisting of individual stories and narratives that influence their fears and uncertainties. Indeed, individuals could vastly overestimate the frequency with which cases of rare adverse vaccine-related effects occur due to a cognitive bias, promoting vaccine hesitancy [48]. Furthermore, we found that unvaccinated individuals reported that their doubts about vaccination were supported by medical opinions and scientific journals. Murphy and colleagues (2021) reported that hesitant individuals were less likely to obtain information about vaccination from traditional, authoritative, and scientific sources, suggesting that scientific value could be used to endorse their choice not to be vaccinated [49]. On the one hand, a higher percentage of unvaccinated individuals reported that the distrust of pharmaceutical companies, healthcare workers, and scientific research influenced their attitudes toward vaccines. On the other hand, a higher number of vaccinated individuals reported trust in healthcare workers, scientific research, and pharmaceutical companies. These findings confirmed that vaccine acceptance involves trusting the providers (e.g., specific healthcare professionals and doctors) and policymakers (e.g., the health system and public researchers) [13,34,50]. Additionally, unvaccinated individuals reported that mass media and newspapers fueled their doubts about vaccination. In this regard, previous studies identified confusing information from mass media [19], the perception of being inadequately informed, and the incomprehensibility of information available [34] as the main reasons for refusing COVID-19 vaccines. Furthermore, unvaccinated individuals mentioned the type of vaccine as a factor influencing their attitudes toward vaccines. In this respect, the more rapid development and approval of the vaccine against COVID-19 compared to previous vaccines against other diseases might have alarmed individuals, generating uncertainties about the vaccine’s safety and efficacy and then becoming common reasons for vaccination refusal [34]. Our study also showed that the decision-making experience of vaccination reflects the interaction of several factors at the individual (i.e., cognitions and emotions), group (i.e., experiences of family members, type of job), and societal level (i.e., misinformation, mistrust in science and government). Our findings are in line with previous studies from Italy and other countries [10,36,37,42], showing that a lack of confidence in vaccinations, distrust in the scientific community and authorities as well as concerns about vaccine-related side effects and safety may decrease vaccination rates.

Overall, the results of this study can help scholars and practitioners delineate the profiles of COVID-19-vaccinated and unvaccinated subjects, clarifying the peculiarities specific to each of these two groups. Gaining a deeper knowledge of the key factors characterizing psychological and cognitive features of vaccine-hesitant/resistant individuals may provide public health officials with relevant information to effectively design and deliver public health messages [19,51]. These findings suggest that successful vaccination campaigns need proactive communication strategies to inform the population and address vaccine concerns. To this end, communication strategies should be improved by providing citizens with adequate and scientific information about vaccines, including possible side effects to help them make a realistic assessment of the costs and benefits of vaccinations. To prevent citizens from relying on dubious sources of information, it could also be helpful to empower citizens to critically assess the accuracy of the information and ensure effective and more accessible science communication on vaccinations. This will allow scholars and practitioners to reach a broader range of citizens and galvanize public support for vaccination as the most cost-effective public health measure to fight diseases. To this end, documents adequately translated by trusted messengers could be disseminated by leveraging the potential of social media platforms. Moreover, the gap between science and society could be bridged by organizing interactive events dedicated to popular science and fun learning on vaccination, including hands-on experiments, science shows, guided visits to research labs, science quizzes, and games.

### Limitations

These findings should be interpreted in light of several limitations. Firstly, these results cannot be considered representative of the entire Italian population and might be biased by the snowball sampling method. However, we believe the large sample size helped us provide accurate and reliable results on the factors influencing vaccination decisions. Secondly, this study merely relies on online self-reported measures, and thus it suffers from the limitations of such a methodology (e.g., social desirability). Additionally, our data were collected using a form from a spreadsheet in Google Sheets, while semi-structured telephone and face-to-face interviews would allow us to extend our data collection to individuals with different digital literacy levels. Thus, future studies should collect longitudinal multimethod data from multiple sources and use means to reach non-Internet users.

Longitudinal research is needed to understand whether the perceptions of vaccinated and unvaccinated individuals can change over time as a function of becoming infected by the virus. Replications on larger and more nationally representative samples are needed to increase the generalizability of the findings of the present study. Thirdly, perceptions of vaccines were qualitatively explored using ad hoc items that were specifically created for this study. Moreover, we adopted a genuine naïve perspective to describe and categorize participants’ responses according to the meaning of words. Thus, we recommend that future qualitative investigations better explore the preliminary findings that emerged from this study in a more structured way. Finally, some information that could be relevant to better understanding vaccine-related perceptions is missing as we deliberately kept the survey short to reduce the response burden and participants’ drop-out rates. Further studies are needed to explore whether and how some unmeasured but potentially relevant factors (e.g., whether participants had previously contracted COVID-19 infection or the number of doses of vaccine received) impact vaccine beliefs and intentions.

## 5. Conclusions

By exploring in-depth perceptions of vaccination among the Italian population, this qualitative study revealed that while vaccinated individuals described vaccination using the ‘safety’, ‘progress’, and ‘ethics’ themes, unvaccinated individuals were more likely to associate vaccination with ‘mistrust’ and ‘ambivalence’ themes. As a result, the latter were more likely to report high-risk perceptions and distrust of the scientific community, pharmaceutical companies, and healthcare providers. Conversely, trust in healthcare, scientific research, and pharmaceutical companies, working in the healthcare sector, and younger age were factors shaping positive perceptions of vaccination and reducing hesitance attitudes. Governments, health policymakers, and media sources should make a joint effort to manage vaccine hesitancy and infodemics and promote vaccination trust among citizens by giving them a forum in which they can make their voices heard and have their concerns about vaccination addressed. To this end, citizens should be provided with prompt communications conveyed through new media and traditionally trusted information channels [31]. Moreover, general practitioners can play a key role in correcting individuals’ erroneous beliefs via counselling and in-depth information actions, which can activate a process of empowerment in local populations [19]. More research is needed to investigate which factors can make an information campaign designed to reduce mistrust and ambivalence toward vaccination a success.

## Figures and Tables

**Figure 1 vaccines-11-00642-f001:**
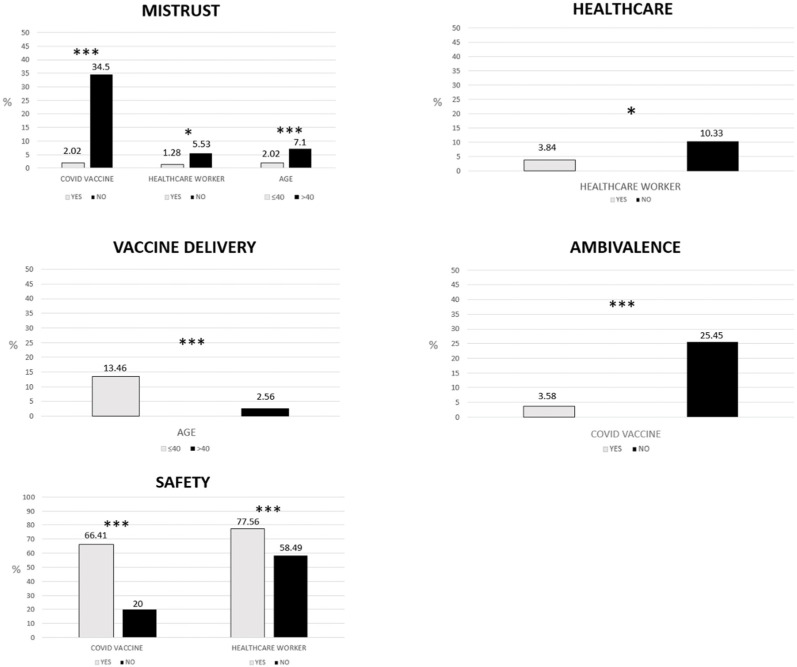
Statistically significant differences in the prevalence of themes. Only statistically significant differences are shown. * *p* < 0.05, *** *p* < 0.001. Comparisons were tested with chi-square. Fisher’s test was performed for categories with *N* ≤ 5.

**Table 1 vaccines-11-00642-t001:** Survey tool.

Items	Response
Socio-Demographic Data	
Age	(Number)
Gender	○ Male ○ Female
Do you work as a healthcare professional?	○ Yes ○ No
Perception of Vaccines	
(a) Which image/word/phrase does come to mind when you think of vaccines?	(Free answer)
(b) Why do you think getting vaccinated may be risky? You can select more than one option.	○ I do not think vaccines are a risk at all ○ Personal negative experience ○ Negative experience of family and/or friends ○ Negative reported experience of acquaintances ○ Medical opinions ○ Media (TV, Internet, etc.), the press ○ Scientific journals
(c) Which of the following variables do you think impact more on your attitudes toward vaccination? You can select more than one option.	○ Type of vaccine (e.g., mRNA, inactivated live virus, etc.). ○ Mode of administration (needle versus spray) ○ Confidence in pharmaceutical companies ○ Distrust of pharmaceutical companies ○ Confidence in healthcare personnel and physicians ○ Distrust of healthcare personnel and physicians ○ Confidence in scientific research ○ Distrust of scientific research ○ Opinions of friends or relatives ○ Mass media, newspapers, magazines ○ Other: …
(d) I have had COVID-19 vaccine.	○ YES ○ NO

**Table 2 vaccines-11-00642-t002:** Frequency of categories in the total sample.

Themes	*n* (%)	Examples
Safety	439 (62.7)	‘Salvation’; ‘Protective Shield’; ‘Needs’
Healthcare	62 (8.9)	‘COVID’; ‘Virus’; ‘Doctor’
Vaccine Delivery	55 (7.9)	‘Injection’; ‘Syringe’; ‘Needle’
Progress	42 (6)	‘Evolution’; ‘Scientific Research’; ‘Future’
Ambivalence	37 (5.3)	‘Uncertainty’; ‘Some safe others not’; ‘Fear’
Mistrust	32 (4.6)	‘Obligation in the form of blackmail’; ‘Fraud, censorship, truth-hiding and drugs for the healthy’; ‘Gene therapy and genotoxicity’.
Ethics	22 (3.1)	‘An act of responsibility’; ‘I vaccinate myself to help others’; ’Civic sense’
Unclassifiable	11 (1.6)	‘Cow’s milk’; ‘To cows’; ‘Daje’

**Table 3 vaccines-11-00642-t003:** Differences in risk perception and factors influencing attitudes toward vaccination between the vaccinated against COVID-19 and the unvaccinated individuals.

	Vaccinated against COVID-19	Unvaccinated against COVID-19	χ^2^	*p*
	*N* (%)	*N* (%)		
Risk perceptions				
No risk	317 (49.15)	2 (3.64)	40.50	<0.001
Negative experience of relatives and/or friends	55 (8.53)	26 (47.27)	70.61	<0.001
Negative experiences reported by acquaintances	80 (12.40)	14 (25.45)	6.35	<0.05
Medical opinions	104 (16.12)	21 (38.18)	15.34	<0.001
Scientific journals	107 (16.59)	29 (52.73)	40.00	<0.001
Attitudes				
Vaccine type	200 (31.01)	42 (76.36)	44.11	<0.001
Trust in pharmaceutical companies	100 (5.12)	1 (1.82)	6.62	<0.05
Distrust of pharmaceutical companies	70 (15.52)	1 (1.82)	164.77	<0.001
Trust in healthcare worker and doctors	258 (40)	2 (3.64)	27.17	<0.001
Mass media, newspapers, magazines	39 (6.05)	12 (21.82)	16.40	<0.001
Distrust of scientific research	21 (3.26)	12 (21.82)	34.85	<0.001
Trust in scientific research	547 (84.81)	9 (16.36)	141.13	<0.001
Distrust of health workers and doctors	18 (2.79)	21 (38.18)	114.02	<0.001

Note. Only statistically significant differences are reported.

## Data Availability

The data that are presented in this study are available on request from the corresponding author. The data are not publicly available due to privacy.

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
