# Peer review of "Perceptions of COVID-19 Vaccines: Protective Shields or Threatening Risks? A Descriptive Exploratory Study among the Italian Population"

_vaccines, 2023, doi:10.3390/vaccines11030642_

Round 1

Reviewer 1 Report

This descriptive qualitative investigation represents a study that includes a sample of the Italian population (total 700 participants who completed an online survey; the inclusion criteria for this study were: to be native Italian speakers and to be at least 18 ys old - mean age of 41 ys). The authors had three main goals: to gain insight into the general population's perception of vaccination; to assess the impact of three specific factors such as anti-SARS-CoV-2 vaccine, employment in the health system and the age of the respondents - on the viewpoint of vaccination and to assess the perception of factors that influence attitudes towards vaccination - especially among vaccinated and among non-vaccinated respondents. This study confirmed that vaccinated persons reported words related to the "safety" theme significantly more often than in unvaccinated group (66% vs. 20%). Opposite, unvaccinated individuals reported words related to the "ambivalence" theme and words related to the mistrust theme more frequently than vaccinated (25% vs. 4% and 35% vs. 2%, respectively). Vaccinated individuals reported that vaccination is associated with "safety", "progress" and "ethics" themes, while unvaccinated group described the vaccination mainly in terms of "mistrust" and "ambivalence". Authors concluded and suggested that communication approaches should be improved by providing sufficient information concerning the vaccines - together with benefits and risks/side effects. The obtained findings indicate the potential usefulness of the joint efforts of governments, health policy actors and different media sources.

I believe that the work is original and represents a novelty in the study of anti-SARS-CoV-2 vaccination field, as well as could be published. However, I suggest that the "Conclusion" section be more concise and significantly shorter.

Reviewer 2 Report

The manuscript submitted by Boragno and co-workers shows the perceptions of COVID-19 Vaccines: Protective Shields or Threatening Risks? A Qualitative Study among the Italian Population.  However, the manuscript has significant faults that the authors should clarify before a new peer-revision round. Some of my comments are below:

·       In my opinion, we can not confirm theses type of study with qualitative parameters.

·       Only limited number of parameters were included to validate the perceptions of COVID-19 vaccines.

·       Data is collected mainly on the basis of google form.

·       The manuscript only describes the obtained results but does not compare with the literature or propose a possible mechanism of action.

·       The conclusions are not supported by the results shown in the manuscript.

The manuscript is not suitable for publication; it should be carefully revised before the following peer-review process.

Reviewer 3 Report

Referee Report for “Perceptions of COVID-19 Vaccines: Protective Shields or Threatening Risks? A Qualitative Study among the Italian Population”

The paper examines the perceptions of COVID-19 vaccines among the Italian population using a qualitative approach. The sample for the study includes 700 participants and is collected online. The study finds out that the vaccinated individuals more frequently used words related to “safety” while the unvaccinated individuals more frequently used words related to “mistrust.” The authors argue that being younger than 40 years old and working in the health sector affected the perceptions in terms of pro-vax attitudes. Finally, the study found that the unvaccinated individuals were more likely to be affected by negative experiences related to their acquaintances and showed higher degrees of mistrust of scientific research, doctors, and pharmaceutical companies.

Comments:

1- There are many paragraphs that has one or a few sentences, which is not a good writing style.

2- The authors argue that although there are other related studies their paper is the first qualitative study in the topic. So, it is a good idea to compare existing studies’ methods with theirs.

3- Of course, it is not possible to re-run the survey at the moment, for future studies, the authors may want to consider the possibility of vaccinated people changing their minds. I have personally seen non-negligible number of people who got vaccinated initially but then stopped getting boosters at some point due to loss in trust. Initially, they fear from COVID19 and get vaccinated but then after seeing some bad examples, they change their mind and don’t get additional vaccinations. So, I suspect that some of the people who got vaccinated might have a change of heart. The number of doses may have been an additional and useful information.

Reviewer 4 Report

In case I missed it,  what were the perceptions of those who remained unvaccinated and were infected? The conclusions are obvious that those who did not get the vaccine did not trust it, so the findings are not additive to what is already known. If the data on those refused the vaccine and get Covid-19 would be very helpful, if available. Similar data should be available for those who received the vaccine, yet still got infected. Did their view? change?

Round 2

Reviewer 2 Report

Authors have justified all the queries and incorporated all the suggestions in the revised version, thus it can  now be considered for publication.

Reviewer 4 Report

The authors have attended to the changes suggested